# Antibacterial and Antifouling Efficiency of Essential Oils-Loaded Electrospun Polyvinylidene Difluoride Membranes

**DOI:** 10.3390/ijms24010423

**Published:** 2022-12-27

**Authors:** Lucie Bartošová, Jana Sedlaříková, Petra Peer, Magda Janalíková, Pavel Pleva

**Affiliations:** 1Department of Environmental Protection Engineering, Faculty of Technology, Tomas Bata University in Zlin, Vavreckova 275, 760 01 Zlin, Czech Republic; 2Department of Fat, Surfactant and Cosmetics Technology, Faculty of Technology, Tomas Bata University in Zlin, Vavreckova 275, 760 01 Zlin, Czech Republic

**Keywords:** PVDF, nanofiber, essential oil, antibacterial activity, antifouling activity

## Abstract

Nanofibers have become a promising material in many industries in recent years, mainly due to their various properties. The only disadvantage of nanofibers as a potential filtration membrane is their short life due to clogging by bacteria in water treatment. The enrichment of nanofibers with active molecules could prevent these negative effects, represented by essential oils components such as Thymol, Eugenol, Linalool, Cinnamaldehyde and Carvacrol. Our study deals with the preparation of electrospun polyvinylidene difluoride (PVDF)-based nanofibers with incorporated essential oils, their characterization, testing their antibacterial properties and the evaluation of biofilm formation on the membrane surface. The study of the nanofibers’ morphology points to the nanofibers’ diverse fiber diameters ranging from 570 to 900 nm. Besides that, the nanofibers were detected as hydrophobic material with wettability over 130°. The satisfactory results of PVDF membranes were observed in nanofibers enriched with Thymol and Eugenol that showed their antifouling activity against the tested bacteria *Escherichia coli* ATCC 25922 and *Staphylococcus aureus* ATCC 25923. Therefore, these PVDF membranes could find potential applications as filtration membranes in healthcare or the environment.

## 1. Introduction

Lately, there has been significant development of nanomaterials worldwide, including many interesting components with excellent physical and chemical properties. These materials are nanoparticles, quantum dots, nanowires, nanotubes, nanorods and nanofibers [1]. The increasing importance of nanomaterials throughout different industries is due to their size, which gives them specific properties such as high surface reactivity. Nanofibers especially, whose diameters are below 1 µm, are experiencing a significant expansion among nanomaterials [2]. They also dispose of remarkable properties, including an exceptionally high surface-to-volume ratio, high porosity and morphology [3]. Moreover, these properties play a crucial role in advanced nanofiber applications in composites for various scientific industries, such as biomedical engineering, healthcare and the environment [1].

However, the choice of polymer for the composite is significant; it plays a crucial role in nanofiber utilization [4]. Among the various polymer materials, polyvinylidene difluoride (PVDF) represents a suitable choice for the environmental industry. It is an inert thermoplastic fluoropolymer with excellent tolerance to many solvents. Moreover, the polymer is water-insoluble, thus the applications of PVDF are widespread in the environment [5,6]. PVDF nanofibers can be used to produce membranes that are fundamentally used for microfiltration, ultrafiltration, membrane distillation, gas separation, removal of pollutants from water, biofuel regeneration, ion exchange processes and other methods [7].

Nevertheless, the use of PVDF membranes is limited by two main problems: contamination during filtration and wetting. Clogging and wetting will reduce the membrane efficiency and its performance, which will increase operating costs and even cause failure. The solution to this issue may be an appropriate hydrophilic or hydrophobic treatment of PVDF membranes [8].

New functional properties of the membranes can be achieved by the incorporation of other substances into the nanofiber structure, especially antibacterial properties that can limit membrane clogging. Scientists started to load active substances into the polymer matrix to enhance the final membrane with functional properties decades ago. To prevent the fast expiration, PVDF membranes commonly used in water treatment, for example, have the antibacterial substance implemented into the structure [8].

Many studies have been dealing with incorporating different kinds of substances, such as fatty acids (FA), monoglycerides (MAG), proteins, types of nutrition and many others [9,10,11,12,13]. These substances have antibacterial efficiency, but they could change the final morphology of nanofibers due to the influence on the polymer solution. However, previous research has overlooked the antibacterial activity of essential oils (EOs) loaded into the PVDF membrane [14]. It was found that even low concentrations of EOs have a great potential as antimicrobial substances. Due to the diverse composition of EOs with antimicrobial properties, these oils can act against a wide range of microorganisms, as there can be a synergistic effect between them [15].

Our study deals with the production of nanofiber membranes based on PVDF polymers, which are used primarily for filtration. The PVDF membranes with incorporated EOs should prevent the membrane from clogging through bacteria aggregation on the membrane surface.

## 2. Results

### 2.1. Physical Characterization of PVDF Solutions

The physical properties of polymer solutions can influence the process of electrospinning. Their viscosity, electrical conductivity and surface tension are significant factors for nanofiber production [16].

Table 1 shows the measured values for viscosity, electrical conductivity, and surface tension for all sample solutions.

The viscosity of PVDF solutions enriched with EOs was slightly lower than those without the EOs, which was probably caused by the addition of the EOs in liquid form. Per Fortunato et al., the rheological analysis of the PVDF/DMF solution demonstrated a Newtonian behavior up to 30% *w*/*w* concentration [17].

As was stated above, viscosity is a crucial factor for electrospinning and can influence the diameter range of nanofibers. The viscosity of the PVDF solution is (3724.6 ± 2.0) mPa·s and it might be predicted that the diameters of the nanofibers from this solution may take high values.

However, the electrical conductivity of the samples was quite diverse. The PVDF solution with the conductivity of (9.90 ± 0.08) μS/cm has the highest value among all samples. The incorporation of the EOs decreased the resulting conductivity. The higher concentration of EOs in the solution, the lower the ability to induce an electric current [18].

In addition, the surface activity of the PVDF solution was measured. According to the results given in Table 2, the surface tension for all the PVDF solutions was almost identical and fluctuated around 38.5 mN/m, which is two times lower than water [19].

### 2.2. Antibacterial Activity of PVDF Films—Disk Diffusion Method

PVDF solutions modified with EOs were used for the preparation of solvent cast films that were solid, flexible and homogenous with white and yellow coloring. Their antibacterial activity was determined by disk diffusion method against Gram-negative bacteria *E. coli* and Gram-positive bacteria *S. aureus.* Inhibition zones were detected immediately, and after 3 and 6 months to reflect the EOs behavior loaded in nanofibers.

Table 2 shows the inhibition zone values in the presence of PVDF films. As can be seen, not all the tested EOs have antibacterial properties. The lower concentration of Lin, Eug and Thy could not stop the bacterial growth of *E. coli* and *S. aureus*. Lin especially has the lowest antibacterial activity of all samples. Even though inhibition zones were observed at concentrations 3% and 5% *w*/*w*, the zones were not as significant for other tested EOs. The best results were provided by 5% *w*/*w* Thy against *E. coli* and *S. aureus* since the zones were over 40 mm in diameter. However, the efficiency of the EOs is different against Gram-negative and Gram-positive bacteria. Eugenol has higher antibacterial activity against *E. coli* because the zones were two times bigger than those against *S. aureus.*

In all cases, the antibacterial effect of the EOs decreased over time, as the inhibition zones have lower values after 3 and 6 months after film preparation. Due to this fact, the theory of evaporation of the EOs from the films over time has been confirmed [20]. Moreover, films loaded with 1% Car, 3% and 5% Lin largely evaporated in three months, and no inhibition was observed.

It can be concluded that Table 2 provides valuable results for assessing the antibacterial activity of polymer films and the stability of incorporated EOs. With a few exceptions, bacterial growth was visibly suppressed in the presence of prepared polymer films. *E. coli* showed the highest sensitivity to the polymer film with the highest concentration (5% *w*/*w*) of EOs. Overall, PVDF films with Thy and Car showed the highest antibacterial efficiency of all tested polymer films.

### 2.3. Morphology of PVDF Nanofibers

After observing the antibacterial activity of EOs loaded into PVDF films, identical nanofibers were produced by electrospinning. Table 3 shows the diameter values of the prepared nanofibers. As can be seen, the values differ substantially and they are from 570 to 900 nm, i.e., they still fall within the criterion for nanofibers. Those with incorporated EOs acquired smaller fiber diameters than pure PVDF nanofibers: (850 ± 50) nm, except nanofibers with 5% *w*/*w* Thy, 3% *w*/*w* Cin and 5% *w*/*w* Eug. This could be caused by polymer solutions with active substances that achieved different viscous properties in comparison to the pure PVDF solution. As described previously, the process of electrospinning is affected by many parameters, especially the viscosity of the polymer solution, which can rapidly change the morphology of the nanofibers [21].

In addition to the smaller diameters of PVDF nanofibers with incorporated EO active substances, various deformations of these fibers were observed in the structure, which was affected by the volatile substances incorporated; this could occur when the solvent evaporated together with the active substances. In addition, it is observed that the diameters are highly variable due to various factors influencing the structure of the nanofibers during the spinning process [22].

Figure 1 shows SEM images for PVDF nanofibers loaded with 5% *w*/*w* EOs and pure PVDF nanofibers. A random arrangement of fibers in many layers can be observed. The fiber diameters of pure PVDF are visibly more extensive than others, except for nanofibers with 5% *w*/*w* Eug. The structure of pure PVDF nanofiber is formed by a thin network with a different distribution of diameters of individual fibers.

Conversely, samples with incorporated EOs formed a denser network of fibers, with obvious structure deformations. A defect can be seen on nanofibers with Lin. In addition, Car nanofibers braided several threads together. In contrast, the Eug nanofibers are formed by a very dense network of fibers with large diameters of individual fibers, and the wavy orientation. Changes in fibrous structure after the incorporation of active molecules were observed also in the study of Peer et al., in which the PVDF-co-HFP based nanofibrous membranes were investigated [23].

### 2.4. The Wettability of PVDF Nanofibers

A hydrophobicity measurement was performed to better understand the material’s interaction with aqueous substances [24]. Furthermore, the surface behavior can influence the adhesion of water-based molecules and organisms and, in our case, biofilm formation.

From the result in Table 4, it is clear that these nanofibers are highly hydrophobic. This observation was not unexpected due to the insolubility of the polymer itself in water. Even the incorporation of EO has not entirely changed the contact angle values compared to pure PVDF. Nevertheless, nanofibers enriched with Lin slightly increased the hydrophobicity of the material. On the contrary, Eug decreased the wettability of the material. Contact angle values fluctuated, but it is not that significant.

### 2.5. Antibacterial Activity of PVDF Nanofibers—Disk Diffusion Method

Similar to PVDF films, the presence of an inhibition zone around the nanofiber membrane disk was evaluated. PVDF nanofibers with incorporated EOs alone proved the presence of these substances via sensory odor evaluation. However, no inhibition zone was confirmed in any sample on the tested bacteria. The absence of inhibition can be attributed to different structure and less surface contact of the nanofibers than in the case of the film. The nanofibers have smaller sizes, and they are distributed in many layers so that the fiber itself comes into contact with the agar surface only on a small surface. Following the results of the disk diffusion method, the nanofibers were subsequently tested to prevent biofilm formation.

### 2.6. Fluorescence Microscopy of PVFD Nanofibers

Since no inhibition was observed around the disk, fluorescence microscopy was performed on individual nanofibers to confirm the presence of sensitivity of the tested bacteria toward the samples. Countless living cells were observed on the surface of pure PVDF (control). Therefore, it was confirmed that the PVDF polymer itself does not exhibit antibacterial properties. Per the images of the PVDF nanofibers enhanced with EOs, it can be concluded that these nanofibers have a bacteriostatic to bactericidal effect. No bacterial cells were observed on the surface of 5% *w*/*w* Eug; the bacteriostatic effect was confirmed because of the suppressed growth of *E. coli* (see Figure 2). The other nanofibers showed a bactericidal effect in addition to the bacteriostatic effect when dead cells were observed on the surface. Many dead bacterial cells were on the nanofibers with 5% *w*/*w* Lin and Thy incorporated. As a result, it can be concluded that nanofibers with incorporated EOs suppress the growth of model bacteria compared to the control.

Similar results were also achieved with disks in the presence of *S. aureus*. The only difference was in the higher number of *S. aureus* dead bacterial cells on the surface of individual nanofibers. According to these results, it can be concluded that the Gram-negative bacteria *S. aureus* is more sensitive to PVDF nanofibers than the Gram-negative bacteria *E. coli*.

### 2.7. Antifouling Activity of PVDF Nanofibers

Biofilm formation was evaluated using SEM images, a high concentration of bacterial cells confirming the presence of biofilm formation. SEM analysis revealed the occurrence of biofilm only on PVDF membranes with EOs concentration 5% *w*/*w*. Figure 3 shows individual SEM images for Gram-positive *S. aureus*. Aggregation of bacterial cells is observed on the PVDF nanofibers without EOs, indicating the biofilm’s presence. The same conclusion was reached for the PVDF nanofibers with incorporated Cin, where densely grouped *S. aureus* cells can be seen. Nanofibers enriched with Car and Cin showed lower cell aggregation than pure PVDF nanofibers.

Conversely, membranes with Thy and Eug have been evaluated as materials that can suppress biofilm formation. As can be seen in Figure 2, the nanofibers with incorporated Thy and Eug—especially Eug—prevent the membrane from bacterial aggregation. These results are very promising for the potential usage of the membranes.

Further, Table 5 shows the positive or negative biofilm formation of *S. aureus* and *E. coli* in the presence of PVDF nanofibers with 5% *w*/*w* EOs. As can be seen, pure PVDF nanofibers provide a comfortable surface for bacterial aggregation, the same as Cin nanofibers. In contrast, loading with 5% *w*/*w* Thy created unfavorable conditions for bacterial formation in the presence of both *E. coli* and *S. aureus*. Similar results were obtained by nanofibers loaded with Eug, but low biofilm formation was observed in the presence of *E. coli*.

Regarding the antifouling activity of PVDF nanofibers, it can be concluded that nanofibers with incorporated Thy and Eug could be potentially used for the production of a filtration membrane. Even though these nanofibers could not stop the bacteria growth completely, they prevent them from aggregation (biofilm formation).

## 3. Discussion

Nanofiber PVDF membranes are widely used, especially in wastewater treatment, but their lifetime and durability are threatened by bacteria that can impede the membrane by aggregating it on the surface. Since the PVDF polymer creates suitable conditions for bacteria and their adhesion, it is necessary to adjust the chemical composition of the membrane. Suitable substances are essential oils or their active compounds with antibacterial properties widely used in biomedical applications.

Representatives of essential oils, Thymol, Linalool, Carvacrol, Cinnamaldehyde and Eugenol, were tested in this study. They were added to the polymer solution in various concentrations, solvent-cast and dried until a film formed. It is known that the properties of nanofibrous membranes are strongly affected by the character of the polymer solution. Thus, the surface tension, conductivity and viscosity of PVDF solutions were measured to complement the complex characteristics of investigated PVDF systems. Solution viscosity depends on the polymer character, its molecular weight and concentration. Control solution based on 23% *w*/*w* PVDF in DMF exhibited the viscosity of 3725 mPa·s, a comparable value with the result of previous study [23,25,26]. PVDF samples modified with EOs’ active compounds exhibited statistically significant different viscosity values that ranged from 2790 to 3725 mPa·s according to the specific formulation. As can be seen in Table 1, viscosity decreased with increasing the addition of active molecules in most cases. The results of conductivity measurement provided the highest value of 9.9 µS/cm for the control solution followed by the decrease in modified PVDF samples. As for the surface tension analysis, relatively low values (around 38.5 mN/m) were obtained regardless of the control or modified PVDF solutions. Despite the facts from the literature, no significant effect of polymer solutions’ properties on the character of the resultant nanofibrous membrane was confirmed by our data. Similarly, as in the previous study, the predominant factor of the applied solvent DMF and it surface tension value is supposed [25,27,28].

The solvent-cast films were subsequently tested for inhibition against *Escherichia coli* and *Staphylococcus aureus* model bacteria using a disk diffusion method. The individual results (Table 2) show that Gram-negative bacteria are more sensitive to these substances than Gram-positive bacteria. The same outcome was obtained by Romeo et al. in their study [29]. Excellent antibacterial results were observed in all samples with 5% *w*/*w* EOs when a significant zone of inhibition was determined. Conversely, samples with 1% *w*/*w* Thy and Lin showed no inhibition compared to other EOs. However, Ibrahim et al. discovered excellent inhibition results against the tested Thy at a concentration of up to 1.5% *w*/*w* [30]. While few studies exist examining the antibacterial activity of pure Lin, high-linalool EOs inhibit the growth of Gram-positive and Gram-negative bacteria [31].

As positive inhibition results with selected EOs were confirmed, nanofibers enriched with these antibacterial agents were prepared using electrospinning and were characterized. Their diameters ranged from 570 to 900 nm. Similar results were obtained in the study of collagen nanofibers with incorporated thyme and oregano EOs, where different fiber diameters and the deformations formed on them were also observed [32]. EOs caused a higher incidence of anomalies in nanofibers’ overall structure than our results when only one active substance was incorporated into the nanofiber. The more significant defects can be attributed to the EOs’ various components, which differ in physical and chemical properties. In the article by Mele et al. dealing with the study of spinning essential oils (especially cinnamon oil) into synthetic polymers, the diameters of nanofibers were smaller (300 ± 60) nm than the experimentally measured values of PVDF nanofibers [33]. It follows that the selection of polymers for electrospinning depends very much on the overall morphology of the nanofiber layer and the resulting required properties.

Besides the morphology, the wettability of the prepared membranes was investigated as it is a significant material property, especially for determination, antifouling and antibacterial activity. The nanofibers were detected as hydrophobic, with the values ranging between 130 to 159°. Their hydrophobic property prevents adhesion of water-based molecules and organisms.

After observing the morphology and wettability of the prepared fibers, they were also tested using a disk diffusion method for their antibacterial activity. However, no inhibition was observed. In the studies of Berechet et al. and Peer et al., they also did not notice an inhibition zone after spinning the polymer solution with the incorporated antibacterial agent [32,34]. In contrast, in the study by Mele et al., the inhibition of nanofibers with incorporated essential oils (oregano, lavender, thyme) in which a needle system spun the individual nanofiber layers was visible in *E. coli* and *S. aureus* [35]. Liu et al. tested the antimicrobial activity of nanofiber membranes with incorporated thyme EO and also noted positive inhibition against the tested bacteria [36]. Their results can be attributed to different electrospinning techniques.

Even though the disk diffusion method of PVDF nanofibers did not report valuable results (no visible inhibition zone), the sensitivity of the model bacteria was demonstrated by fluorescence microscopy. The results showed lower presence of live bacterial cells on the surface of PVDF nanofibers enhanced with EOs than on the control. In addition, these samples were also tested to evaluate antifouling activity. Biofilm formation was suppressed by nanofibers with incorporated 5% *w*/*w* Thy and Eug, where no aggregation of bacterial cells was observed on the surface of the nanofibers in both tested bacteria. Alves Carniero et al. dealt with EOs’ behavior as antimicrobials and anti-biofilm substances of *S. aureus* [36]. They found that EOs containing a large proportion of the active substance Thy prevented biofilm formation, the same as in this study. However, further studies are needed to better understand the interactions between biofilm formation steps in the presence of EOs and their components.

Although satisfactory results in suppressing biofilm formation have been achieved for PVDF membranes with incorporated Thy and Eug—and thus these membranes could find application as potential filtration membranes—further optimizations are needed for the production of nanofibers with antibacterial properties. The biggest obstacle is the incorporated substance itself, which is most likely to evaporate with the solvent during the spinning process and is not incorporated in a concentrated form.

## 4. Materials and Methods

### 4.1. Materials and Chemicals

Polyvinylidene fluoride Kynar^®^ (PVDF) was purchased from Arkema (Colombes, France). As a solvent, N, N′-dimethylformamide p.a. (DMF) purchased from VWR Chemicals (France) was used. Active compounds of essential oils (EOs) with potential antibacterial properties were chosen: Thymol (Thy), Carvacrol (Car), Linalool (Lin), *trans*-Cinnamaldehyde (Cin) and Eugenol (Eug). These compounds were purchased from Sigma-Aldrich (USA) and used without further purification.

Model bacterial strains chosen for testing antibacterial properties were Gram-negative bacteria *Escherichia coli* ATCC 25922 and Gram-positive bacteria *Staphylococcus aureus* ATCC 25923. These bacterial strains were cultivated on Mueller Hinton agar (MHA, HiMedia, Mumbai, India) and Mueller Hinton broth (MH, HiMedia, Mumbai, India) at 37 °C in a thermostat.

### 4.2. Preparation of Polymer Solutions

A PVDF polymer solution was prepared at concentration 23% *w*/*w* and dissolved in an organic solvent DMF using a magnetic stirrer Heildoph (Schwabach, Switzerland) (250 rpm, 40 °C, 30 min). Subsequently, three different concentrations of EOs’ active compounds (Thy, Car, Lin, Cin, Eug) were added into polymer solutions to their final concentrations of 1% *w*/*w*, 3% *w*/*w* and 5% *w*/*w*.

These compounds were quantitatively added to homogeneous, clear, viscous PVDF solutions and stirred on a magnetic stirrer Heidolph (250 rpm, 25 °C, 30 min). Sixteen PVDF solutions were prepared.

### 4.3. Preparation of Polymer Films

The polymer films were prepared as control of antibacterial activity of incorporated EOs in the polymer before the nanofiber production.

After stirring, 5 g of each prepared polymer solution was cast into glass Petri dishes with a diameter of 60 mm, which were then dried at 40 °C for 24 h to evaporate solvent gradually. As a result, solid homogenous polymer films were formed, stored at 60% relative humidity and 22 °C ± 1 °C for up to 6 months. The storage of the films was observed to better understand the EOs evaporation during the time.

### 4.4. Physical Characterization of PVDF Solutions

Due to the incorporation of EOs compounds, the physical characterization of the PVDF solutions was determined to better understand EOs’ influence on electrospun nanofibers and their further behavior. The viscosity, electrical conductivity and surface tension were observed on these samples.

The viscosity was measured by Brookfield DV—III Ultra Rheometer Ametek (Brookfield, Middleboro, MA, USA) at a shear rate of 20 rpm with spindle LV3. All measurements were determined at 26.2 °C ± 0.4 °C.

The electrical conductivity was tested by WTW inoLab^®^ Cond 7100 (WTW, Prague, Czech Republic) at 25.6 °C ± 0.7 °C.

The surface tension of the PVDF solutions was also analyzed. The samples were measured by KRUSS Easy Dyne Tensiometer (KRUSS, Matthews, NC, USA) at 26.1 °C ± 0.5 °C.

All the measurements were carried out in triplicate.

### 4.5. Preparation of Nanofibers

Electrospinning with a needleless system was used to produce nanofibers from the PVDF solutions given above. The nanofiber layers were prepared on a laboratory device which consisted of a high voltage source Spellman SL70PN150 (Hauppage, New York, NY, USA), a 10 mm diameter metal rod and a stationary flat grounded collector with aluminum foil (the device was originally built in the Department of Hydrodynamics, Czech Academy of Science, Prague). According to previous studies [37], PVDF solutions were spun at a distance of 150 mm from the collector under voltage of 18 kV and defined environmental conditions (temperature 24.0 °C ± 0.2 °C, relative humidity 48% ± 2%). The process consisted of transferring 0.2 mL of PVDF solution on the metal rod, then the electrospinning was performed for 4 min on aluminum substrates.

### 4.6. Morphology of PVDF Nanofibers

The structure of the prepared samples of nanofibers (shown in Table 1) was observed by scanning electron microscope (SEM) TESCAN VEGA 3 (Tescan, Brno, Czech Republic). The samples (2 × 2 mm) were coated with a conductive gold layer using a QUORUM Q150R ES magnetron sputtering device (Quorum, Hertfordshire, United Kingdom). The morphology was characterized after placing the coated samples in SEM.

### 4.7. The Wettability of PVDF Nanofibers

The surface properties of PVDF nanofibers were studied using the sessile droplet method on Attension Theta tensiometer (Biolin Scientific, Västra Frölunda, Sweden) in combination with OneAttension software at ambient conditions. Distilled water was used as the reference liquid with a droplet volume of 3 µL. All the measurements were tested in triplicate.

### 4.8. Antibacterial Activity—Disk Diffusion Method

The antibacterial activity of the prepared polymer films and nanofibers was tested on *Escherichia coli* and *Staphylococcus aureus* using the disk diffusion method. First, disks of polymer films and nanofibers with a diameter of 9 mm were cut. Bacterial suspensions with 0.5 McFarland turbidity in saline solution were prepared from 24-h bacterial cultures incubated in MH broth (Mumbai, India). The prepared bacterial suspension was pipetted in a volume of 1 mL on the entire surface of MH agar. The excess suspension was pipetted off and the prepared disks of polymer films and nanofibers were placed in the Petri dishes. It was followed by a thermostat culturing the Petri dishes with samples at 37 °C for 24 h. After incubation, antibacterial efficiency was evaluated in the form of inhibition zones. This method was repeated after 3 and 6 months after the first experiment, each in triplicate.

### 4.9. Fluorescence Microscopy

This method was performed only on PVDF nanofibers because of the inconvenient results from the disk diffusion method. Disks were removed from the Petri dish and placed on a glass slide. This was followed by dyeing them with fluorescent dye (SYTO^®^9 and propidium iodide) for 10 s and then covering them with a square coverslip. The fluorescence microscope Olympus BX53 (Olympus, Tokyo, Japan) equipped with Microscope Digital Camera DP73 (Olympus, Tokyo, Japan) and the cell Sens Standard 1.18 (Olympus, Tokyo, Japan) software was used to determine this method. The live and dead bacteria cells were absorbed according to the dye’s interaction with the cell membrane.

### 4.10. Antifouling Activity of PVDF Nanofibers

Biofilm formation was tested in glass tubes with 3 mL of BHI broth (Brain Heart Infusion, Himedia, Mumbai, India) + 5% *w*/*w* sucrose (Himedia, Mumbai, India) and 60 μL of a 0.5 McFarland turbidity bacterial suspension with the sample of the PVDF nanofiber disk incubated at 37 °C for 72 h. After incubation, nanofiber samples were rinsed thoroughly from planktonic cells with distilled water. The antifouling activity was determined by the fluorescence microscope OLYMPUS DP73 and scanning electron microscope TESCAN VEGA 3.

### 4.11. Statistical Analysis

Data from the characterization of PVDF and antimicrobial activity tests (disk and well diffusion method) were expressed as mean ± standard deviation (SD). Statistical analysis was carried out by a one-way ANOVA followed by a Tukey test using Statistica software version 10, StatSoft, Inc. (Tulsa, OK, USA) at the significance level of *p* < 0.05. 

## 5. Conclusions

During the experiment, PVDF-based nanofiber membranes enhanced with active compounds of EOs, namely Thy, Eug, Lin, Car and Cin at different concentrations, were successfully prepared by electrospinning. Additionally, the physical characterization of the PVDF solutions were determined to obtain better information about their impact on the nanofiber production. The fiber diameters of the individual nanofiber membranes were very diverse in how they were affected by the incorporation of EOs. The individual fibers showed a circular cross-section, a random arrangement and, in some cases, defects in the form of droplets and clumps of fibers. The prepared hydrophobic nanofibers were then subjected to antibacterial tests, such as the disk diffusion method and fluorescence microscopy. Even though no inhibition zone was observed in the presence of PVDF nanofibers, the results from the microscopy were valuable. The results show bacteriostatic and even bacteriocidic activity of EO-enhanced nanofibers. In addition to antibacterial activity, the biofilm formation on the surface of the PVDF membrane was assessed. In the case of *S. aureus*, biofilm formation was not confirmed in membranes with 5% *w*/*w* Eug and Thy. These results are positive for further research, focusing on a new approach to incorporating EOs into the nanofiber structure to achieve antibacterial properties. According to the results of antifouling activity, these membranes have great potential as filter membranes in healthcare or the environment in terms of applicability.

## Figures and Tables

**Figure 1 ijms-24-00423-f001:**
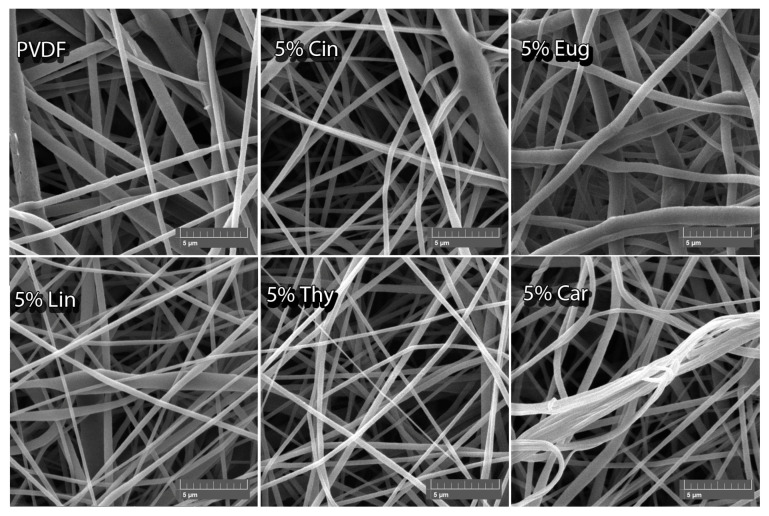
SEM images of PVDF pure nanofibers and ones loaded with 5% *w*/*w* of individual EOs.

**Figure 2 ijms-24-00423-f002:**
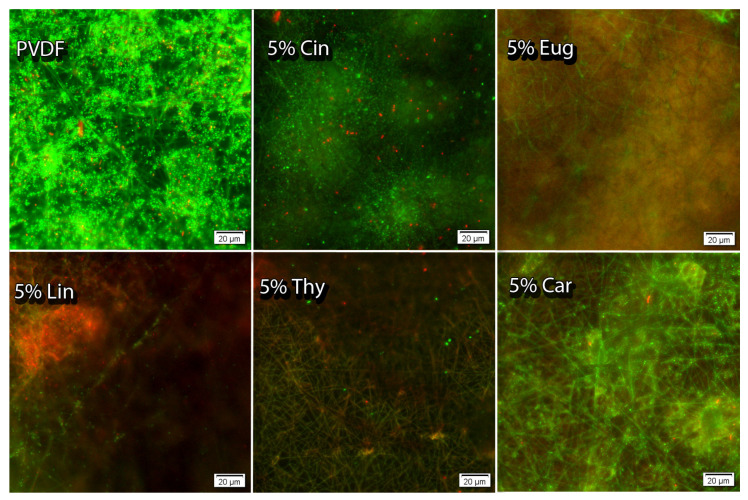
The viability of *E. coli* presence of PVDF nanofibers with EOs (5 *w*/*w*).

**Figure 3 ijms-24-00423-f003:**
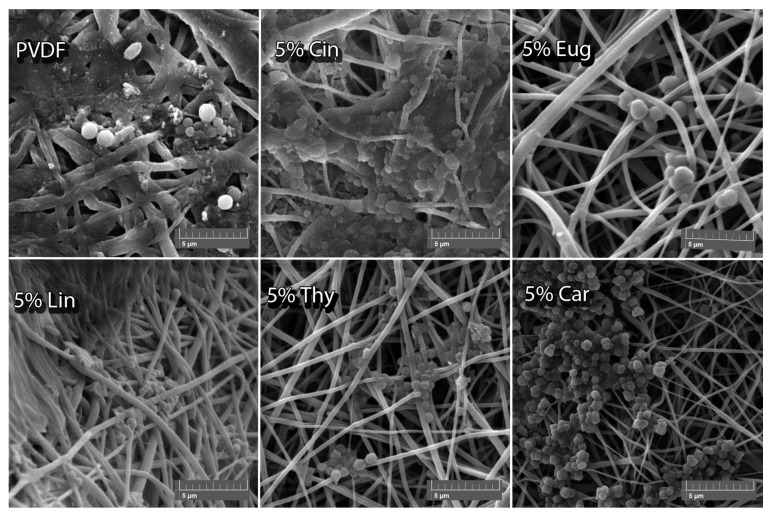
Biofilm formation SEM images of pure PVDF nanofibers and with individual EOs in the presence of *S. aureus*.

**Table 1 ijms-24-00423-t001:** Physical characterization of PVDF solutions (viscosity, electrical conductivity and surface tension).

23% PVDF +	*w*/*w* [%]	Viscosity [mPa·s]	Electrical Conductivity [μS/cm]	Surface Tension [mN/m]
Pure PVDF (control)	-	3724.6 ± 2.0 ^a^	9.90 ± 0.08 ^a^	38.51 ± 0.51 ^a,b,c^
Cin	1	3387.5 ± 1.0 ^b^	9.43 ± 0.05 ^b^	38.70 ± 0.53 ^a,b,c^
	3	2790.6 ± 7.0 ^c^	8.73 ± 0.05 ^c^	38.80 ± 0.30 ^b,c^
	5	3316.1 ± 3.0 ^d^	7.10 ± 0.07 ^d^	38.92 ± 0.60 ^b,c^
Eug	1	3944.8 ± 4.0 ^e^	9.47 ± 0.05 ^b^	39.60 ± 0.62 ^c^
	3	3517.7 ± 2.0 ^f^	8.87 ± 0.05 ^c,e^	39.11 ± 0.33 ^b,c^
	5	3409.7 ± 1.1 ^g^	8.47 ± 0.05 ^f^	38.90 ± 0.44 ^b,c^
Lin	1	3302.3 ± 2.0 ^h^	9.63 ± 0.05 ^g^	38.34 ± 0.55 ^a,b^
	3	3394.7 ± 0.7 ^i^	8.97 ± 0.05 ^e^	38.41 ± 0.59 ^a,b,c^
	5	2993.0 ± 0.3 ^j^	8.50 ± 0.08 ^f^	37.70 ± 0.51 ^a^
Thy	1	3719.2 ± 1.0 ^k^	9.67 ± 0.05 ^g^	38.00 ± 0.25 ^a^
	3	3116.3 ± 1.1 ^l^	8.90 ± 0.08 ^c,e^	38.60 ± 0.56 ^a,b,c^
	5	3337.1 ± 1.0 ^m^	8.37 ± 0.05 ^f^	38.82 ± 0.60 ^a,b,c^
Car	1	3610.0 ± 5.0 ^n^	9.57 ± 0.05 ^b,g^	38.31 ± 0.53 ^a,b^
	3	3521.9 ± 1.8 ^f^	8.97 ± 0.05 ^e^	38.40 ± 0.59 ^a,b,c^
	5	2875.8 ± 1.0 ^o^	8.50 ± 0.08 ^f^	38.70 ± 0.51 ^a,b,c^

Lowercase letters in the columns denote significant differences (*p* < 0.05).

**Table 2 ijms-24-00423-t002:** Inhibition zone values of PVDF films for bacteria *E. coli* and *S. aureus* (disk diameter—9 mm).

*E. coli*				*S. aureus*		
	Inhibition Zone [mm]			Inhibition Zone [mm]		
PVDF + EOs (*w*/*w*)	Promptly	After 3 Months	After 6 Months	Promptly	After 3 Months	After 6 Months
Pure PVDF (control)	x ^a^	x ^a^	x ^a^	x ^a^	x ^a^	x ^a^
1% Cin	14.2 ± 0.2 ^d^	9.7 ± 0.3 ^a^	x ^a^	13.6 ± 0.5 ^c^	9.6 ± 0.3 ^a,b^	x ^a^
3% Cin	25.0 ± 0.5 ^g,h^	24.5 ± 0.6 ^g^	13.8 ± 0.6 ^d^	30.2 ± 0.5 ^h,i^	25.0 ± 0.4 ^f^	13.2 ± 0.3 ^c^
5% Cin	35.7 ± 0.3 ^j^	35.7 ± 0.4 ^j^	27.0 ± 0.6 ^i^	31.6 ± 0.3 ^i^	30.0 ± 0.4 ^h^	30.2 ± 0.5 ^h,i^
1% Eug	x ^a^	x ^a^	x ^a^	x ^a^	x ^a^	x ^a^
3% Eug	24.5 ± 0.3 ^g^	21.7 ± 0.3 ^f^	18.5 ± 0.6 ^e^	10.2 ± 0.3 ^b^	9.6 ± 0.3 ^a,b^	10.2 ± 0.6 ^b,c^
5% Eug	34.7 ± 0.3 ^j^	25.8 ± 0.2 ^h^	24.7 ± 0.3 ^g^	16.2 ± 0.5 ^d^	12.6 ± 0.3 ^c^	12.8 ± 0.9 ^c^
1% Lin	x ^a^	x ^a^	x ^a^	x ^a^	x ^a^	x ^a^
3% Lin	9.7 ± 0.3 ^a^	x ^a^	x ^a^	9.8 ± 0.3 ^b^	x ^a^	x ^a^
5% Lin	11.8 ± 0.2 ^b,c^	x ^a^	x ^a^	10.2 ± 0.3 ^b^	x ^a^	x ^a^
1% Thy	x ^a^	x ^a^	x ^a^	x ^a^	x ^a^	x ^a^
3% Thy	24.5 ± 0.3 ^g^	24.0 ± 0.6 ^g^	14.8 ± 0.4 ^d^	31.6 ± 0.3 ^i^	25.6 ± 0.5 ^f^	15.4 ± 0.3 ^d^
5% Thy	41.7 ± 0.7 ^k^	40.7 ± 0.6 ^k^	35.8 ± 0.5 ^j^	40.8 ± 0.5 ^l^	40.4 ± 0.3 ^l^	34.8 ± 0.6 ^j^
1% Car	12.0 ± 0.6 ^b^	x ^a^	x ^a^	15.4 ± 0.3 ^d^	x ^a^	x ^a^
3% Car	21.2 ± 0.4 ^f^	19.7 ± 0.3 ^e^	10.3 ± 0.3 ^a,b^	24.4 ± 0.3 ^f^	20.0 ± 0.4 ^e^	12.6 ± 2.0 ^c^
5% Car	34.3 ± 0.3 ^j^	33.7 ± 0.8 ^j^	25.8 ± 0.6 ^g,h,i^	37.2 ± 0.3 ^k^	35.4 ± 0.5 ^j^	27.4 ± 0.5 ^g^

x—no inhibition zone was detected. Lowercase letter in the columns denote significant differences (*p* < 0.05).

**Table 3 ijms-24-00423-t003:** Diameters of PVDF nanofibers.

	*w*/*w* [%]	d [nm]
Pure PVDF (control)	-	850 ± 50
Thy	1	770 ± 30
	3	900 ± 60
	5	590 ± 40
Car	1	600 ± 40
	3	610 ± 40
	5	640 ± 30
Cin	1	700 ± 60
	3	570 ± 30
	5	600 ± 30
Lin	1	710 ± 40
	3	700 ± 60
	5	650 ± 30
Eug	1	800 ± 50
	3	570 ± 30
	5	890 ± 50

**Table 4 ijms-24-00423-t004:** Contact angle values of PVDF nanofibers.

	*w*/*w* [%]	Contact Angle [°]
Pure PVDF (control)	-	142.4 ± 1.1 ^a,b^
Cin	1	145.1 ± 1.1 ^b^
	3	158.5 ± 1.2 ^c^
	5	148.9 ± 0.9 ^d^
Eug	1	130.1 ± 1.5 ^e^
	3	134.0 ± 3.0 ^e^
	5	138.5 ± 1.5 ^a^
Lin	1	143.2 ± 0.9 ^b^
	3	152.0 ± 3.0 ^d^
	5	151.2 ± 1.7 ^d^
Thy	1	147.6 ± 0.9 ^d,f^
	3	140.5 ± 0.7 ^a^
	5	137.6 ± 1.8 ^a,e^
Car	1	148.8 ± 0.4 ^d^
	3	145.1 ± 1.5 ^b,f^
	5	147.0 ± 0.7 ^f^

Lowercase letter in the columns denote significant differences (*p* < 0.05).

**Table 5 ijms-24-00423-t005:** Biofilm formation in the presence of PVDF nanofibers.

23% *w*/*w* PVDF Nanofibers	*E. coli*	*S. aureus*
Pure PVDF (control)	++ *	++
5% Eug	+	-
5% Lin	+	+
5% Cin	++	++
5% Car	++	+
5% Thy	-	-

* ++ strong biofilm formation, + low biofilm formation, - no biofilm formation.

## Data Availability

Not applicable.

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
