# Peer review of "Antibacterial and Antifouling Efficiency of Essential Oils-Loaded Electrospun Polyvinylidene Difluoride Membranes"

_ijms, 2022, doi:10.3390/ijms24010423_

Round 1
Reviewer 1 Report
The presented manuscript, entitled "Antibacterial and Antifouling Efficiency of Electrospun Polyvinylidene difluoride Membranes with Incorporated Essential Oils’ Compounds," offers an exciting and important study on nanofiber membranes based on PVDF, which are used primarily for filtration. The PVDF membranes with incorporated essential oils should prevent the membrane from clogging by bacteria aggregation on the membrane surface.
I have some comments and suggestions for improvement of the manuscript:
1. In this study, you also produce films along with fibrous membranes. Is there a particular reason for this? Please state why did you produce both films and membranes. In the objectives, state what you want to achieve?
2. I recommend dividing paragraph 2.2 on:
2.2. Preparation of polymer solutions.
Subsequently,
2.3. Physical characterization of solutions,
2.4. Preparation of films and
2.5. preparation of nanofibers.
In its current form, it isn't very clear. Since there is no explicit statement in the introduction that both films and membranes will be made, the preparation of films may surprise the reader.
3. Please, describe why you stored the films in the controlled environment for up to 6 months.
4. How do you explain the fluctuation in the viscosities of solutions? One would expect the viscosity to decrease or increase as the additive is added. However, the expected trend for solutions with Cin and Thy is not observed.
5. All facts from lines 185 - 196 are known from the literature. Subsequently, the discussion also lacks a treatise on the influence of viscosity, conductivity, and surface tension on the morphology of the fibers in this specific case. Why were conductivity and surface tension measured?
6. Why was wettability studied? Does wettability affect bacteriostatic, bactericidal, antibacterial, and antifouling activity? Please bring these contexts into the discussion.
7. In line 292, there is some typos e a.
8. Please, check the manuscript; there are typos like disc or disk, etc.
Author Response
Dear Editor and Reviewers,
Thank you for giving me the opportunity to submit a revised draft of our manuscript titled Antibacterial and Antifouling Efficiency of Essential Oils-Loaded Electrospun Polyvinylidene difluoride Membranes to the Special Issue “SMART and Macromolecular Biomaterials, from Materials to Biology”. We appreciate the time and effort that you and the reviewers have dedicated to providing your valuable feedback on our manuscript. We are grateful to the reviewers for their insightful comments on our paper. We have been able to incorporate changes to reflect all the suggestions provided by the reviewers.
We look forward to hearing from you in due time regarding our submission and to respond to any further questions and comments you may have.
Sincerely,
Pavel Pleva et al.
(corresponding author)
Tomas Bata University in Zlin
Faculty of Technology
Department of Environmental and Protection Engineering
Vavreckova 275
760 01 Zlin
Email: [email protected]
Here is a point-by-point response to the reviewers’ comments and concerns.
Comment 1: In this study, you also produce films along with fibrous membranes. Is there a particular reason for this? Please state why did you produce both films and membranes. In the objectives, state what you want to achieve?
Response 1: Thank you for pointing this out. The films were produced to be more confident about the antibacterial properties of incorporated EOs into the polymer. Results of the disc diffusion method was the base for further nanofibers’ preparation and characterization. The observation of the films also delivered satisfactory results about the EOs' behaviour through their storage under defined conditions, as the inhibition zones decreased throughout time, which can reflect the behaviour of these compounds in the nanofibers.
Comment 2: I recommend dividing paragraph 2.2 on:
2.2. Preparation of polymer solutions.
Subsequently,
2.3. Physical characterization of solutions,
2.4. Preparation of films and
2.5. preparation of nanofibers.
In its current form, it isn't very clear. Since there is no explicit statement in the introduction that both films and membranes will be made, the preparation of films may surprise the reader.
Response 2: Thank you for your suggestions. All the authors agreed with your suggested division of the paragraphs to be more understandable for the readers. They were changed in the manuscript.
Comment 3: Please, describe why you stored the films in the controlled environment for up to 6 months.
Response 3: Thank you for your comment. As was previously mentioned in Response 1, the films were prepared firstly to predict the antibacterial properties of incorporated EOs. Since the outcomes from the disc diffusion method of nanofibers did not bring satisfactory data as in case of the films, we decided to store the films under controlled conditions to observe the EOs‘ behaviour in time. As the applications of produced nanofibers can be diverse, for example, as potential SMART material (packaging, filtration membrane, …), the evaporation of the EOs from the material is significant for further operations. This simple observation gave us some point of view on how EOs could possibly be released from nanofibers in time. We also included an explanation of the storage of the films in the manuscript.
Comment 4: How do you explain the fluctuation in the viscosities of solutions? One would expect the viscosity to decrease or increase as the additive is added. However, the expected trend for solutions with Cin and Thy is not observed.
Response 4: Thank you for your comment. We expect that viscosity fluctuation could be ascribed to different type of interaction of polymer chain and bioactive molecules at their various concentrations. The molecular structure of added substances plays an important role, too. On the other hand, a theoretical presumption of viscosity decrease was shown in all the samples when the lowest and highest added concentrations of active molecule is considered. Also, a decreasing trend in viscosity of modified samples was confirmed in comparison to control PVDF solution.
Comment 5: All facts from lines 185 - 196 are known from the literature. Subsequently, the discussion also lacks a treatise on the influence of viscosity, conductivity, and surface tension on the morphology of the fibers in this specific case. Why were conductivity and surface tension measured?
Response 5: Thank you for pointing this out. The measurement of conductivity, viscosity and surface tension was included in the study to complement the complex characteristics of investigated PVDF systems, despite the fact that no significant effect of polymer solutions' properties on the character of resultant nanofibrous membrane was confirmed by our data, contrary to literature facts.
Comment 6: Why was wettability studied? Does wettability affect bacteriostatic, bactericidal, antibacterial, and antifouling activity? Please bring these contexts into the discussion.
Response 6: Thank you for your comment. Wettability is a significant material property for determination, for example, its antibacterial and antifouling efficiency. It is related to the adhesion of the bacterial cells to the material. In our case, the nanofibers were detected as hydrophobic; the material prevents from adhesion of water-based molecules and organisms.
Commenr 7: In line 292, there is some typos e a.
Response 7: Thank you for pointing this out. The typos were corrected in manuscript.
Comment 8: Please, check the manuscript; there are typos like disc or disk, etc.
Response 8: Thank you very much. We checked the manuscript and corrected the typos in it.
Reviewer 2 Report
The manuscript reports the preparation of oil-loaded PVDF nanofibers and their antibacterial and antifouling applications. The contents are interesting and fall within the scope of IJMS, I recommend its acceptance for publication after the following issues are well addressed.
The title can be more concise such as “ Electrospun Essential oil-loaded Polyvnylidene difluoride Membranes for Antibacterial and Antifouling applications”
It should be better to include key quantitative data results in your abstract for the readers to cite your job.
Recent progresses about electrospun PVDF nanofibers (https://doi.org/10.3390/polym14204311; 10.3390/ijms232214322) can be recommended to the readers.
Several sentences about the most recent developments of electrospinning can be added to project one of the merits in your article, e.g. electrospinning can be exploited to encapsulate all kinds of molecules, including little chemical molecules (10.3390/ijms23105444; https://doi.org/10.3390/polym14224947), drugs (10.3390/ijms23137147; https://doi.org/10.3390/polym13244286), proteins and peptides (10.2147/IJN.S370340; 10.3390/ijms232214055), nutritions (10.1007/s42114-022-00551-x), and also nanoparticles (https://doi.org/10.3390/biom12091254) in monolithic (10.3390/polym12102421), core-shell (https://doi.org/10.3390/membranes11110802), tri-layer core-shell (10.1016/j.jallcom.2020.156471) and other complicated nanostructures (10.3389/fchem.2022.944428). However, limited efforts have been pay to the oils loaded nanofibers and their potential applications (10.3390/pharmaceutics14061208). Your job is just a kind example in this direction.
How about DMAc or DMSO to replace DMF for preparing the working fluids?
Please pay attention to the Significant figures, SUCH AS 38.7 ± 0.51 please be 38.70 ± 0.51 in Table 1. The same format for all the data.
In the discussion, a key issue please be discussed, which is about the efficiency of encapsulation of oil in the nanofibers, how much has been escaped to the environment during the electrospinning process.
Please unify your references’ formats, e.g. the up and lower case in the articles’ titles.
Author Response
Dear Editor and Reviewers,
Thank you for giving me the opportunity to submit a revised draft of our manuscript titled Antibacterial and Antifouling Efficiency of Essential Oils-Loaded Electrospun Polyvinylidene difluoride Membranes to the Special Issue “SMART and Macromolecular Biomaterials, from Materials to Biology”. We appreciate the time and effort that you and the reviewers have dedicated to providing your valuable feedback on our manuscript. We are grateful to the reviewers for their insightful comments on our paper. We have been able to incorporate changes to reflect all the suggestions provided by the reviewers.
We look forward to hearing from you in due time regarding our submission and to respond to any further questions and comments you may have.
Sincerely,
Pavel Pleva et al.
(corresponding author)
Tomas Bata University in Zlin
Faculty of Technology
Department of Environmental and Protection Engineering
Vavreckova 275
760 01 Zlin
Email: [email protected]
Here is a point-by-point response to the reviewers’ comments and concerns.
Comment 1: The title can be more concise such as “ Electrospun Essential oil-loaded Polyvnylidene difluoride Membranes for Antibacterial and Antifouling applications”
Response 1: Thank you for your suggestion. We rewrote the title to be more attractive to the readers.
Comment 2: It should be better to include key quantitative data results in your abstract for the readers to cite your job.
Response 2: Thank you very much for pointing this out. The key quantitative data were included in the abstract.
Comment 3: Recent progresses about electrospun PVDF nanofibers (https://doi.org/10.3390/polym14204311; 10.3390/ijms232214322) can be recommended to the readers. Several sentences about the most recent developments of electrospinning can be added to project one of the merits in your article, e.g. electrospinning can be exploited to encapsulate all kinds of molecules, including little chemical molecules (10.3390/ijms23105444; https://doi.org/10.3390/polym14224947), drugs (10.3390/ijms23137147; https://doi.org/10.3390/polym13244286), proteins and peptides (10.2147/IJN.S370340; 10.3390/ijms232214055), nutritions (10.1007/s42114-022-00551-x), and also nanoparticles (https://doi.org/10.3390/biom12091254) in monolithic (10.3390/polym12102421), core-shell (https://doi.org/10.3390/membranes11110802), tri-layer core-shell (10.1016/j.jallcom.2020.156471) and other complicated nanostructures (10.3389/fchem.2022.944428). However, limited efforts have been pay to the oils loaded nanofibers and their potential applications (10.3390/pharmaceutics14061208). Your job is just a kind example in this direction.
Response 3: Thank you for your suggestion. We decided to expand the introduction by other possible molecules’ incorporations into PVDF nanofibers.
Comment 4: How about DMAc or DMSO to replace DMF for preparing the working fluids?
Response: 4: Thank you for your comment. There are numerous solvents and their mixtures that can be used. For simplicity, we chose DMF as in our previous study (10.1021/acsami.1c07257).
Comment 5: Please pay attention to the Significant figures, SUCH AS 38.7 ± 0.51 please be 38.70 ± 0.51 in Table 1. The same format for all the data.
Response 5: Thank you for the notice. The significant data were corrected throughout the manuscript.
Comment 6: In the discussion, a key issue please be discussed, which is about the efficiency of encapsulation of oil in the nanofibers, how much has been escaped to the environment during the electrospinning process.
Response 6: Thank you for pointing this out. As the results of the nanofibers’ antibacterial properties by disc diffusion method were not the same as the film results, we decided to find out how different the EOs concentration incorporated in films and nanofibers is. The unified weight of films and nanofibers was dissolved in DMF, and the absorbance was measured at a defined wavelength (in triplicate), selected on the basis of calibration for the appropriate active substance. The absorbance results of films and nanofibers gave us information about the EOs incorporated in nanofibers versus in films. The concentrations of EOs in nanofibers were up to 90% less than in films. These results were not included in the manuscript because the method was selected just for comparison of two forms of samples, films and nanofibers with the benefit of convenience and easy implementation.
Comment 7: Please unify your references’ formats, e.g. the up and lower case in the articles’ titles.
Response 7: Thank you. We checked and corrected the references in the manuscript.
Round 2
Reviewer 1 Report
Thank you for the corrections. In this form, the manuscript is suitable for publishing in IJMS.